# Zoledronate in the prevention of Paget's (ZiPP): protocol for a randomised trial of genetic testing and targeted zoledronic acid therapy to prevent *SQSTM1*-mediated Paget's disease of bone

Owen Cronin,[1] Laura Forsyth,[2] Kirsteen Goodman,[3] Steff C Lewis,[2] Catriona Keerie,[2] Allan Walker,[2] Mary Porteous,[4] Roseanne Cetnarskyj,[5] Lakshminarayan R Ranganath,[6] Peter L Selby,[7] Geeta Hampson,[8] Rama Chandra,[9] Shu Ho,[10] Jon H Tobias,[11] Steven Young-Min,[12] Malachi J McKenna,[13] Rachel K Crowley,[13] William D Fraser,[14] Luigi Gennari,[15] Ranuccio Nuti,[15] Maria Luisa Brandi,[16] Javier Del Pino-Montes,[17] Jean-Pierre Devogelaer,[18] Anne Durnez,[19,20] Giancarlo Isaia,[21] Marco Di Stefano,[21] Núria Guañabens,[22] Josep Blanch,[23] Markus J Seibel,[24,25] John P Walsh,[26,27] Mark A Kotowicz,[28] Geoffrey C Nicholson,[29] Emma L Duncan,[30,31,32] Gabor Major,[33] Anne Horne,[34] Nigel L Gilchrist,[35] Maarten Boers,[36] Gordon D Murray,[37] Keith Charnock,[38] Diana Wilkinson,[38] R Graham G Russell,[39] Stuart H Ralston[4]

For numbered affiliations see end of article.

**Correspondence to**
Dr Stuart H Ralston;
stuart.ralston@ed.ac.uk

## ABSTRACT

**Introduction** Paget's disease of bone (PDB) is characterised by increased and disorganised bone remodelling affecting one or more skeletal sites. Complications include bone pain, deformity, deafness and pathological fractures. Mutations in sequestosome-1 (*SQSTM1*) are strongly associated with the development of PDB. Bisphosphonate therapy can improve bone pain in PDB, but there is no evidence that treatment alters the natural history of PDB or prevents complications. The Zoledronate in the Prevention of Paget's disease trial (ZiPP) will determine if prophylactic therapy with the bisphosphonate zoledronic acid (ZA) can delay or prevent the development of PDB in people who carry *SQSTM1* mutations.

**Methods and analysis** People with a family history of PDB aged >30 years who test positive for *SQSTM1* mutations are eligible to take part. At the baseline visit, participants will be screened for the presence of bone lesions by radionuclide bone scan. Biochemical markers of bone turnover will be measured and questionnaires completed to assess pain, health-related quality of life (HRQoL), anxiety and depression. Participants will be randomised to receive a single intravenous infusion of 5 mg ZA or placebo and followed up annually for between 4 and 8 years at which point baseline assessments will be repeated. The primary endpoint will be new bone lesions assessed by radionuclide bone scan. Secondary endpoints will include changes in biochemical markers of bone turnover, pain, HRQoL, anxiety, depression and PDB-related skeletal events.

**Ethics and dissemination** The study was approved by the Fife and Forth Valley Research Ethics Committee on 22 December 2008 (08/S0501/84). Following completion of the trial, a manuscript will be submitted to a peer-reviewed journal. The results of this trial will inform clinical practice by determining if early intervention with ZA in presymptomatic individuals with *SQSTM1* mutations can prevent or slow the development of bone lesions with an adverse event profile that is acceptable.

**Trial registration number** ISRCTN11616770

### Strengths and limitations of this study

► This is the first randomised placebo-controlled trial to test whether genetic testing coupled with targeted intervention with zoledronic acid can modify the development and progression of bone lesions secondary to Paget's disease.

► The inclusion of individuals with sequestosome-1 mutations ensures that participants have a high risk of developing Paget's disease and provides a suitable cohort in which to study the potential benefit of prophylactic treatment.

► The choice of zoledronic acid maximises the likelihood of preventing the development of bone lesions in this high-risk population.

► The randomised double-blind placebo-controlled design of the trial reduces the risk of selection and assessment bias.

► The study duration is unlikely to be sufficient to evaluate the effect of the intervention on complications of Paget's disease such as bone pain, deformity and pathological fractures. Longer periods of follow-up of this cohort will be required.

## BACKGROUND

Paget's disease of bone (PDB) is characterised by areas of increased and disorganised bone turnover affecting one or more skeletal sites. Many affected individuals develop complications such as bone pain, deformity, deafness, pathological fracture and osteoarthritis. Genetic factors play an important role in PDB. Many genetic variants have been identified that predispose to PDB and related syndromes[1 2] but mutations in sequestosome-1 (*SQSTM1*) are the most important predisposing factor, occurring in up to 50% of patients with a family history of PDB and up to 10% of people who are unaware of having a family history.[3–8] Carriers of *SQSTM1* mutations have more severe disease with an earlier age at onset than patients who do not carry mutations.[9] Cross-sectional studies indicate that the penetrance of PDB in *SQSTM1* mutation carriers is about 80% above the age of 70 years.[10] However, there has been only one prospective study of disease evolution in mutation carriers. This study involved 10 families with *SQSTM1* mutations. Four out of 23 (17%) mutation carriers had developed the PDB disease by the age of 50 as assessed by radionuclide bone scanning, but the age at diagnosis was delayed compared with their parents' age.[11] Bisphosphonates are considered to be the treatment of choice for PDB, and within the bisphosphonates, zoledronic acid (ZA) is most likely to give a favourable response in terms of bone pain.[12] ZA has a long duration of action in patients with established PDB with inhibitory effects on bone turnover that are superior to those of risedronate.[13] The main indication for bisphosphonate treatment in PDB is bone pain thought to be due to increased metabolic activity of the disease.[12 14] Although bisphosphonates are highly effective at suppressing elevated bone turnover in PDB, there is no evidence that giving bisphosphonate treatment with the primary aim of suppressing elevated bone turnover is of clinical benefit.[14] There is also no evidence that treatment can prevent complications of PDB such as fracture, bone deformity, deafness or secondary osteoarthritis.[14 15 16] The Zoledronate in the Prevention of Paget's disease (ZiPP) trial has been designed to determine if prophylactic treatment with ZA is effective at preventing the development or progression of bone lesions with the characteristic features of PDB in asymptomatic *SQSTM1* mutation carriers. The reason for choosing people with *SQSTM1* mutations is that these are relatively common in PDB and because carriers of *SQSTM1* mutations have a high risk of developing PDB with increasing age.

### Good clinical practice

The study will be carried out according to the principles of the International Conference on Harmonisation Tripartite Guideline for Good Clinical Practice and local guidance and regulations

### Consolidated standards of reporting trials

The results of the trial will be reported in accordance with the Consolidated Standards Of Reporting Trials statement.

---

> **Box 1  Outcome measures for the Zoledronate in the Prevention of Paget's disease trial**
>
> **Primary**
> ► The total number of subjects who develop new bone lesions on radionuclide bone scans with the characteristics of Paget's disease of bone (PDB) between the baseline visit and the final follow-up visit.
>
> **Secondary**
> ► The number of new bone lesions on radionuclide bone scan.
> ► Change in activity of existing bone lesions that were present at the baseline assessed by semiquantitative analysis of radionuclide bone scans.[28]
> ► The development of PDB-related skeletal events defined as any one of the following:
> Development of new bone lesions thought to be due to PDB on imaging.
> Development of complications thought to be due to the development or progression of PDB including pathological fractures, bone deformity, deafness, joint replacement surgery or other orthopaedic procedures.
> Administration of treatment for PDB with an antiresorptive drug because of the development of signs or symptoms thought to be due to PDB such as pain localised to an affected site or neurological symptoms.
> ► The development of increased bone turnover as assessed by measurement of biochemical markers of bone resorption and bone formation.
> ► Quality of life, anxiety and depression assessed by the Short Form (36) Health Survey, Brief Pain Inventory (BPI) and Hospital Anxiety and Depression Questionnaire.
> ► Location, presence and severity of pain assessed by the BPI manikin and pain questionnaire.

### Aim

To improve clinical outcome in PDB by exploring whether genetic testing for *SQSTM1* mutations coupled with prophylactic treatment with ZA, can prevent or delay the development of bone lesions, prevent skeletal complications and favourably modify pain and quality of life in those genetically at risk because of a positive family history.

### Objectives

The study objectives are summarised in box 1 and described in more detail below:

#### Primary objective:

To quantify the number of subjects in each treatment group who develop new bone lesions with the characteristics of PDB as assessed by radionuclide bone scan. The presence and location of lesions will be assessed by imaging experts blinded to treatment allocation. A new bone lesion will be defined as evidence of involvement of a new bone or part of a previously affected bone at the end-of-study visit which was not thought to be involved at the baseline visit.

#### *Secondary objectives:*

To quantify the number of new bone lesions and change in activity of existing bone lesions present at baseline; to

**Table 1** Summary of assessments and outcome measures for the ZiPP trial

| | Screening | Baseline visit | +1 week | Annual review | End of study |
|---|---|---|---|---|---|
| Medical history | | ✓ | | ✓ | ✓ |
| Current medication | | ✓ | | ✓ | ✓ |
| Physical examination | | ✓ | | | |
| Height, weight, blood pressure | | ✓ | | | ✓ |
| Routine biochemistry* | ✓ | ✓ | | ✓ | ✓ |
| Haematology† | | ✓ | | | ✓ |
| Blood for biomarkers‡ | | ✓ | | | ✓ |
| Urine for biomarkers§ | | ✓ | | | ✓ |
| SQSTM1 genotyping | ✓ | | | | |
| 25(OH) vitamin D | ✓ | | | | |
| Pregnancy test¶ (in women of childbearing potential) | | ✓ | | | |
| Radionuclide bone scan | | ✓ | | | ✓ |
| Radiographs or other imaging** | | ✓ | | | ✓ |
| Infusion | | ✓ | | | |
| Telephone Questionnaire | | | ✓ | | |
| Food Frequency Questionnaire | | ✓ | | | |
| SF-36, HADS and BPI questionnaires | | ✓ | | ✓ | ✓ |
| PDB-related skeletal events | | | | | ✓ |

*Calcium, albumin/total protein, alkaline phosphatase, liver function (AST, ALT, GGT, bilirubin), urea and electrolytes and creatinine.
†Full blood count.
‡Blood samples for measurement of bone-specific alkaline phosphatase, PINP, CTX-I and other specialised markers of bone metabolism.
§Second-voided morning urine to be taken and stored for measurement of N-telopeptide collagen cross links, deoxypyridinoline/creatinine ratio and other specialised markers of bone metabolism.
¶A negative pregnancy test must be obtained on the day of, or the day before, infusion of the study drug. The preferred method of is serum beta-hCG, but a urine beta-hCG is acceptable for centres that are unable to obtain a serum beta-hCG.
**To be taken of relevant areas in subjects suspected to have PDB-like bone lesions on bone scan.
ALT, Alanine aminotransferase; AST, Aspartate aminotransferase; BPI, Brief Pain Inventory; CTX-I, C-terminal telopeptide of type I collagen; GGT, Gamma glutamyl transferase; HADS, Hospital Anxiety and Depression Questionnaire; hCG, Human chorionic gonadotrophin; PDB, Paget's disease of bone; PINP, N-terminal propeptide of type I procollagen; SF-36, Short Form (36) Health Survey ; *SQSTM1*, sequestosome-1; ZiPP, Zoledronic acid in the Prevention of Paget's disease.

evaluate the effects of treatment on skeletal events related to PDB; to evaluate the effects of treatment on biochemical markers of bone resorption and bone formation and to evaluate the effects of treatment on health-related quality of life (HRQoL), anxiety and depression and the presence, localisation and severity of pain.

### Outcome measures

The schedule of assessments and outcome measures, which will be collected during the study, are summarised in table 1 and are discussed individually in more detail below.

### Bone lesions

These will be assessed by Tc[99] radionuclide bone scan, which is recognised to be the most sensitive imaging technique for identifying bone lesions in PDB.[14 17] Participants thought to have PDB-like bone lesions on scan may have further imaging performed by X-ray, CT scan or MRI scan if the local investigator considers it clinically indicated. Anonymised bone scan and X-ray images will be uploaded on to the study database for review. All scans will be reviewed by an imaging expert blinded to treatment allocation. A proportion of images will be reviewed by a second imaging expert, also blinded to treatment allocation, to evaluate concordance between the observers. The images selected will include all of those considered by the primary imaging expert to represent PDB-like lesions. In the event that the experts disagree on a specific image, a third imaging expert (also blinded to treatment allocation) will be asked to adjudicate.

### Clinical assessments

Participants will undergo a physical examination at the baseline visit including blood pressure and pulse. Participants will be evaluated clinically at the end-of-trial visit

for any symptoms or signs of skeletal events thought to be related to PDB.

## Biochemical markers

Measurements of serum creatinine, urea and electrolytes, serum total alkaline phosphatase (ALP), serum calcium, albumin and liver function tests which will consist of aspartate aminotransferase (AST) alanine aminotransferase (ALT), gamma glutamyl transferase (GGT) and bilirubin along with a full blood count will be performed using standard techniques at the local laboratories in participating centres. Estimated glomerular filtration rate (GFR) will be calculated from serum creatinine, gender and weight by the Cockcroft-Gault equation. [18] Specialised biochemical markers of bone turnover will be measured centrally at the University of East Anglia. These will include urine N-telopeptide collagen cross links (NTX) corrected for urinary creatinine; C-terminal telopeptide of type I collagen (CTX-I), bone-specific alkaline phosphatase (BSAP) and the N-terminal propeptide of type I procollagen (PINP). These measurements will be made on fasting samples collected between 09:00 and 12:00 hours as previous studies have shown that markers of bone resorption have a circadian rhythm and are influenced by food intake. [19] The urine samples will be second-voided 'spot' samples collected after an overnight fast. The preferred markers of bone resorption are urinary NTX and serum CTX-I. These have been found to be elevated in patients with PDB in case–control studies [14] and to correlate with the extent of bone lesions as determined by scintigraphy in PDB. [20] The preferred markers of bone formation will be PINP and BSAP since both have been shown to be superior to total ALP at detecting PDB in case–control studies. [14] Additional biomarkers of bone metabolism may also be assessed if new information indicates that these may be of interest as the study progresses. The serum, plasma and urine samples will be aliquoted and stored locally at −80°C and shipped on dry ice to the central laboratory.

## Health-related quality of life

HRQoL will be assessed by completion of the Short Form (36) Health Survey (SF-36) questionnaire [21] at baseline, annual visits and the end-of-study visit. The SF-36 is a widely used, validated questionnaire [21] previously used to assess quality of life in patients with established PDB. [15 16]

## Pain

The presence and location of pain will be assessed by completion of the Brief Pain Inventory (BPI) [22] at baseline, annual visits and the end-of-study visit. The BPI was originally developed to evaluate the location and severity of pain in patients with malignant disease but has since been validated in people with chronic non-malignant pain. [23] In addition to completing BPI, participants will also be asked if they have experienced any pain and bone pain

## Anxiety and depression

Anxiety and depression will be assessed by the Hospital Anxiety and Depression Questionnaire (HADS). [24] This questionnaire was chosen since it is quick and simple to administer and has it has been extensively validated in many different countries and settings. [25]

## Paget's disease-related skeletal events

Participants will be evaluated clinically at the end of study for the presence of Paget's disease-related skeletal events (PDRSE). These will include pathological fractures, bone deformity, deafness due to skull involvement, and joint replacement surgery or other surgical procedures that are a carried out because of PDB. Administration of an antiresorptive drug during the study because of signs or symptoms that are thought to be due to PDB will be considered as a PDRSE as will the development of new bone lesions on bone scan. All events will be combined for each treatment group to give a total score.

## Genetic testing

Genetic testing will be conducted to determine eligibility by Sanger sequencing of exons 7 and 8 of *SQSTM1* and the intron–exon boundaries using DNA extracted from a venous blood sample according to standard techniques. [9]

## Sample size

The sample size was chosen assuming that 15% of patients in the placebo group and 1.5% of patients in the active (ZA) treatment group will develop new PDB-like bone lesions during follow-up. This was based on the observation that ZA has been reported to normalise biochemical markers of bone turnover for up to 6.5 years in 90% of patients with established PDB. [26] With this assumption, 85 subjects in each group would provide 89% power to detect a treatment effect of this magnitude at an alpha of 0.05. Since it is possible that more than one affected subject per family could be enrolled, the sample size was inflated to account for relatedness of individuals. This was done by calculating the mean squared alkaline phosphatase values in patients within families who carried the same mutation (271.3) and the mean squared alkaline phosphatase values between families (619.7) and combining this with the estimated average number of two subjects per family who may be enrolled in the study. This resulted in a design effect factor of 1.39, inflating the required sample size to 118 per group. In addition to this, the sample size was further inflated to account for a 10% rate of participants lost to follow-up resulting in a total sample size of 130 subjects per group or 260 subjects in total. The actual number of subjects randomised to the interventional study by the time recruitment had closed in April 2015 was 222 and to the observational study was 135. The decision to stop recruitment was based on funding, and justified by recalculating the design factor based on the actual number of subjects per family that had been enrolled into the study (1.5 on average). The design factor was recalculated to be 1.26.

**Box 2 Eligibility and exclusion criteria for the Zoledronate in the Prevention of Paget's disease trial**

**Eligibility criteria**
► Carrier of sequestosome-1 mutation.
► Aged 30 years or older.
► Not already diagnosed with Paget's disease of bone (PDB).

**Exclusion criteria**
► Already diagnosed with PDB.
► Unwilling or unable to provide informed consent.
► Contraindication to bisphosphonates.
► Estimated GFR <35 mL/min.
► Hypocalcaemia.
► Receiving bisphosphonate therapy for another reason.
► Osteonecrosis of the jaw.
► Metastatic cancer or cancer diagnosed less than 2 years ago where treatment is still ongoing.
► Active uveitis, iritis or episcleritis.
► Already taking part in another randomised controlled clinical trial.
► Pregnancy or lactation at the time of randomisation or bone scanning.

## Methodology
### Eligibility
Those eligible will be 30 years of age or older, with a positive family history of PDB, in whom genetic testing had shown a pathogenic mutation in *SQSTM1*. Individuals who had already been diagnosed with PDB prior to the baseline visit will be excluded, as will those with contra-indications to ZA as summarised in box 2. In order to identify people who may be eligible for participation, an extensive programme of genetic testing of probands for *SQSTM1* mutations will be carried out. An overview of this process is summarised in figure 1. Patients with a diagnosis of PDB (probands) identified through various sources will be contacted by letter and asked if they would like to be tested for the presence of mutations in the *SQSTM1* gene. Those that test negative for *SQSTM1* mutations will be informed of the result and counselled but they and their family members excluded from further involvement in the study. Those that tested positive will be informed of the result, counselled about the implications and asked to pass an information pack about the trial onto any eligible blood relatives with a reply slip that can be returned to the local recruiting centre. Subsequently, a programme of genetic testing for *SQSTM1* mutations will be conducted on relatives to identify individuals who may be eligible to take part in the trial. The results of this process are summarised in figure 2. Participants with serum 25(OH) vitamin D levels below the lower limit of the local reference range will be permitted to take part in the trial but only after they had been treated with vitamin D supplements in order to mitigate the risk of hypocalcaemia following ZA treatment. Recruitment into

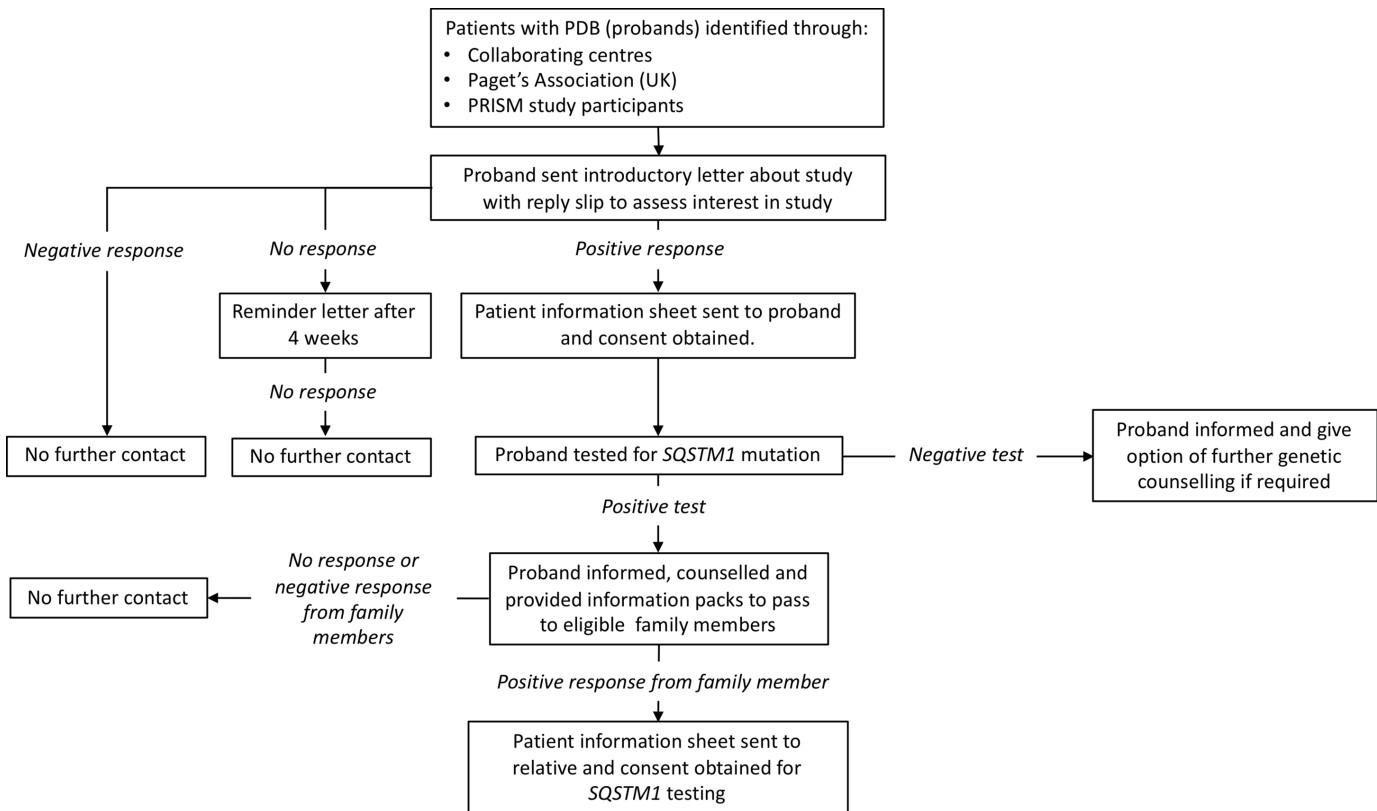

**Figure 1** Genetic testing phase of the ZiPP study for probands the figure provides an overview of the process and procedures for genetic testing of probands and contact of family members. PDB, Paget's disease of bone; PRISM, Paget's Disease: Randomised Trial of Intensive versus Symptomatic Management; *SQSTM1*, sequestosome-1; ZIPP, Zoledronic acid in the Prevention of Paget's disease.

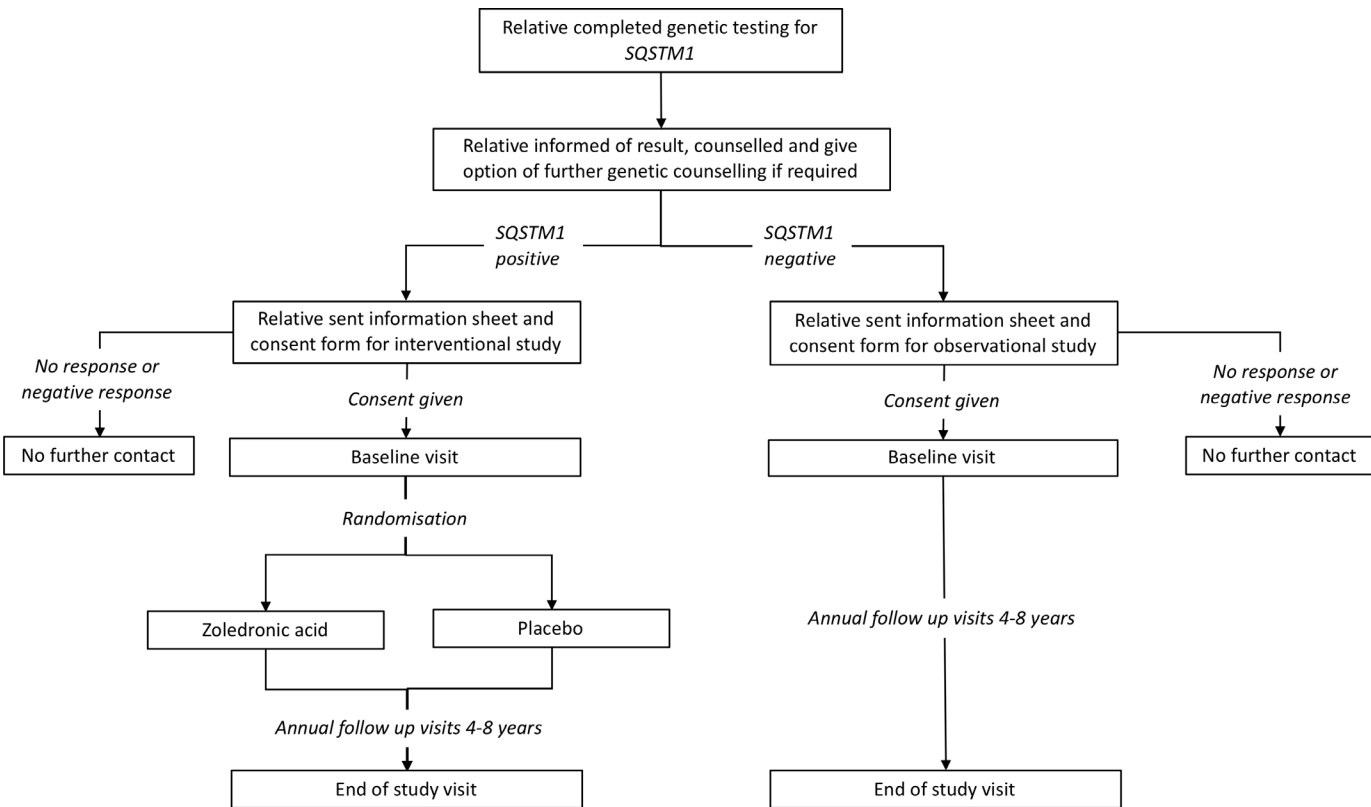

**Figure 2** Genetic testing phase for relatives and subsequent enrolment to the ZiPP study. The figure provides an overview of the process and procedures for genetic testing of relatives as well as an outline of the flow of subjects who consented to participate in the intervention and observational studies. PRISM, Paget's Disease: Randomised Trial of Intensive versus Symptomatic Management; *SQSTM1*, sequestosome-1; ZiPP, Zoledronic acid in the Prevention of Paget's disease.

the clinical trial will be delayed in participants who are scheduled to have dental surgery (tooth extractions, root treatment or other surgery to the mandible or maxilla), until healing had occurred to mitigate the risk of osteonecrosis of the jaw. Likewise, if the potential participant has dental surgery planned within the first 3 months of the expected infusion date, their recruitment will be delayed until healing is complete. Minor dental procedures such as descaling and fillings will not constitute a barrier to enrolment. Participants in Ireland will be required to undergo a dental examination within 1 month prior to the baseline visit at the request of Health Products Regulatory Authority in Ireland. Women who are pregnant or breast feeding will be excluded. Women of childbearing potential will be permitted to take part provided that they agreed to practise a medically robust form of contraception before and for at least 12 months after the ZA infusion (an intrauterine device, a barrier method with spermicide, condoms, subdermal implant or oral contraceptive). In the event that a woman becomes pregnant or is lactating during the study, bone scanning and X-rays will not be performed until the patient is no longer pregnant and has ceased lactating.

### Observational study

During the genetic testing phase, we identified 400 individuals who tested negative for *SQSTM1* mutations and 135 (33.7%) agreed to enrol into an observational study.

Participants in the observational study will have HRQoL and anxiety and depression measures assessed by completion of the SF-36 and HADS questionnaires at the baseline and end-of-study visits. They also will have samples for routine biochemistry checked at baseline and the end-of-study visit and will have samples stored for assessment of biochemical markers of bone turnover.

### Consent

The consent process will be divided into three stages. The first phase will involve obtaining consent from patients with PDB (probands). The second phase will involve obtaining consent from relatives probands for genetic testing. Although the relatives will be made aware of the trial, the consent will be obtained only for genetic testing, without any commitment to enter the trial. The third phase will involve obtaining consent for entry into the trial or observational study.

### Randomisation

Randomisation will be performed by a web-based tool hosted by Edinburgh Clinical Trials Unit, to ensure allocation concealment prior to enrolment. The randomisation algorithm will employ minimisation to ensure that the groups are balanced for prognostic variables thought to influence the occurrence of PDB including: the type of mutation (missense vs truncating or frameshift); gender; whether the baseline radionuclide bone scan shows

lesions suggestive of PDB; whether serum total alkaline phosphatase levels at baseline are elevated (yes/no); and by age band: 30–40, 41–50, 51–60, 61–70 and 71 or over. Following randomisation, the study database will generate a treatment code which will be used by the research pharmacies in each participating centre to ensure that the correct medication is dispensed.

### Prerandomisation and postrandomisation withdrawals
Participants will be advised that they have the right to withdraw from the study at any time for any reason. The investigator will have the right to withdraw a participant at any time if it is deemed to be in the participant's best interest. If a participant decides that they no longer wish to continue with routine assessments or adhere to the study protocol before the planned end-of-trial assessment, they will be given the opportunity to attend for the end-of-trial assessment. The same will apply to participants in whom the local investigator decides that adherence to the trial protocol would be inappropriate.

### Blinding
The participants and investigators will be blinded to treatment allocation. The ZA and placebo infusions will be identical. Breaking the blind will only be performed where knowledge of the treatment is absolutely necessary for further management of the patient and can only be performed by contacting the local pharmacy, which will have restricted code break details.

### Interventions
The investigational medicinal product (IMP), ZA (Aclasta, Novartis Pharmaceuticals UK, Surrey, UK), has been widely used in the treatment of both osteoporosis and PDB.[12 13] The most common side effects are transient influenza-like symptoms occurring in up to 50% of patients although these are usually mild.[27] The IMP will be given by intravenous infusion and will comprise ZA (5 mg in 100 mL ready-to-infuse solution) or a matching placebo. Both will be given at a constant infusion rate over not less than 15 min. Medications required for the participants' clinical care will be permitted during the study. Should a participant require treatment with a bone active antiresorptive medication (such as a bisphosphonate, strontium ranelate or denosumab) after randomisation but prior to infusion of the IMP, then the participant will not receive the study IMP but will still be followed up as per protocol. Female patients of childbearing potential will be required to have a negative pregnancy test on the day of, or the day before, infusions of the study drug. Participants who are sexually active will receive specific advice about the possible risks associated with getting pregnant while in the trial and will be asked to agree to practise a medically acceptable form of birth control for at least 12 months postinfusion of IMP. Female participants who inadvertently become pregnant during the trial will be excluded from isotope bone scanning during pregnancy or breast feeding.

### Data management
Paper case record forms (CRF) will be provided to record baseline and follow-up clinical measurements and demographics by local research teams. Data from these CRF will then be entered onto a web-based electronic CRF. The principal investigator (PI) at each study site will be responsible for the quality of the data recorded in the CRF. The ZiPP study eCRF web portal will be built and maintained by the software development team of the University of Edinburgh's Clinical Trials Unit, following internal standard operating procedures. A Microsoft stack will be used. The back-end repository will be a MS SQL-Server. The front-end user interface will be implemented using ASP.Net technologies.

### Adverse event management
Participants will be provided with an event diary to record details of primary care visits, medications taken, hospitalisations and any other adverse effects or health problems. In the event of hospitalisation, the patient will be asked to contact the PI at their local study centre. Adverse events (AE), serious AEs (SAE) and suspected unexpected serious adverse reactions (SUSAR) will be collected continuously throughout the trial. In addition, participants will be contacted by local research teams 1 week after receipt of the infusion to record symptoms or side effects related to this intervention. All AEs will be recorded from the time a participant consents to join the study until the last study visit has been completed. The investigator or a delegated member of the study team will record AEs at every visit and participants will be instructed to contact the investigator at any time if AEs develop. If an AE/SAE occurs, it is the responsibility of the investigator to review all the documentation related to the event and evaluate seriousness, causality, severity and expectedness. Events that are considered serious, possibly, probably or definitely related to the IMP (serious adverse reactions) and unexpected (SUSAR) may be unblinded if it is necessary for clinical care. Once the investigator becomes aware that an SAE has occurred, they must report the information to the clinical research governance and quality assurance office of the sponsor within 24 hours. The investigator will then be required to complete an SAE form to assess causality, seriousness, severity and expectedness of the event.

## ANALYSIS
### Statistical analysis
The principal analysis will be conducted on an intention-to-treat basis. All analyses will allow for clustering by family, and all primary analyses will be adjusted for the minimisation of variables. Comparisons will be performed using an appropriate linear modelling procedure, taking into account repeated measures where these are available. Patients with completely missing data for a particular outcome will be removed from the analysis of that particular outcome. The effect of this will be examined using

sensitivity analysis. Other sensitivity analyses will look at unadjusted analyses and the effect of adjusting for centre.

## Trial oversight

Monitoring will be performed in accordance with a study monitoring plan developed by the trial's sponsor. The PIs and institutions involved in the study will permit trial-related monitoring, audits, research ethics committee review and regulatory inspection(s). A trial steering committee (TSC) will be established to oversee the conduct and progress of the trial. An independent data monitoring committee will be established to oversee the safety of subjects in the trial. The study is expected to provide new information on the evolution of PDB in this participant group as well to give an indication as to whether ZA treatment can modify the natural history of the disease. Given the relatively short time frame it's unlikely that the trial will demonstrate any clinical benefits of the treatment in terms of complications of PDB such as pain, fractures, deafness or bone deformity, but patients will be evaluated clinically for the presence of any of these complications should they occur.

## Patient and public involvement

The study was designed with the involvement of patients and the Paget's Association—a patient support group. The TSC included a representative of the Paget's Association and a patient representative.

## ETHICS AND DISSEMINATION

The results of the study will be submitted to a peer-reviewed journal so that they are disseminated to the wider medical community. The results will also be disseminated to patients with PDB and their families through the website of the Paget's Association. Authorship on the main paper will be determined by the International Committee of Medical Journal Editors guidelines. The results of the ZiPP trial are expected to inform clinical practice and influence clinical guidelines for PDB by determining if early intervention with ZA in presymptomatic individuals with *SQSTM1* mutations can prevent or slow the development of bone lesions with an AE profile that is acceptable.

## Author affiliations

[1]Department of Rheumatology, Western General Hospital, Edinburgh, UK
[2]Edinburgh Clinical Trials Unit, Usher Institute, University of Edinburgh, Edinburgh, UK
[3]Glasgow Caledonian University School of Nursing Midwifery and Community Health, Glasgow, UK
[4]MRC Institute of Genetics and Molecular Medicine, University of Edinburgh, Edinburgh, UK
[5]University of Dundee - Fife Campus, Kirkcaldy, UK
[6]University of Liverpool, Liverpool, UK
[7]University of Manchester, Manchester, UK
[8]Guy's and St Thomas' NHS Trust, London, UK
[9]King's College Hospital, London, UK
[10]Robert Jones and Agnes Hunt Orthopaedic and District Hospital NHS Trust, Oswestry, UK
[11]Musculoskeletal Research Unit, Translational Health Sciences, Bristol Medical School, University of Bristol, Bristol, UK
[12]Portsmouth Hospitals NHS Trust, Portsmouth, UK
[13]University College Dublin, Dublin, Ireland
[14]Department of Medicine Norwich Medical School, Faculty of Medicine and Health Sciences, University of East Anglia, Norwich, UK
[15]University of Siena Faculty of Medicine and Surgery, Siena, Italy
[16]University Hospital Careggi, Firenze, Italy
[17]University Hospital of Salamanca, Salamanca, Spain
[18]Cliniques Universitaires Saint-Luc, Bruxelles, Belgium
[19]Rheumatology, Cliniques Universitaires Saint-Luc, Bruxelles, Belgium
[20]Algemeen Ziekenhuis Jan Portaels, Vilvoorde, Belgium
[21]Università degli Studi di Torino, Torino, , Italy
[22]Rheumatology Department, Hospital Clinic, IDIBAPS, CIBERehd, University of Barcelona, Barcelona, Spain
[23]Hospital del Mar, Barcelona, , Spain
[24]Concord Repatriation General Hospital, Sydney, New South Wales, Australia
[25]Bone Research Program, ANZAC Research Institute, The University of Sydney, Sydney, New South Wales, Australia
[26]Department of Endocrinology and Diabetes, Sir Charles Gairdner Hospital, Nedlands, Western Australia, Australia
[27]Medical School, The University of Western Australia, Crawley, Western Australia, Australia
[28]Department of Endocrinology and Diabetes, Barwon Health, Geelong, Victoria, Australia
[29]Rural Clinical School, University of Queensland, Toowoomba, Queensland, Australia
[30]Department of Endocrinology and Diabetes, Royal Brisbane & Women's Hospital, Brisbane, Queensland, Australia
[31]Faculty of Medicine, University of Queensland, Herston, Queensland, Australia
[32]Translational Genomics Group, Institute of Health and Biomedical Innovation, Faculty of Health, Queensland University of Technology, Woolloongabba, Queensland, Australia
[33]Royal Newcastle Centre John Hunter Hospital, University of Newcastle, Newcastle, New South Wales, Australia
[34]Department of Medicine, University of Auckland, Auckland, New Zealand
[35]GCM Research Trust, Burnwood Hospital, Christchurch, New Zealand
[36]Epidemiology & Biostatistics, Amsterdam Rheumatology and Immunology Center, Amsterdam University Medical Centers, Vrije Universiteit, Amsterdam, Netherlands
[37]Usher Institute, University of Edinburgh, Edinburgh, United Kingdom
[38]The Paget's Association, Manchester, UK
[39]Mellanby Centre for Bone Research, University of Sheffield, Sheffield, UK

**Acknowledgements** The authors wish to acknowledge the valuable support of the UK Paget's Association in publicising and supporting the study and the many patients with Paget's disease and relatives who supported the study. The authors also wish to acknowledge the contribution of the many research nurses in study centres, Ms Lyndsay Milne from ECTU for data management support, and the laboratory staff in the South East Scotland Genetics Service for conducting mutation analysis of the *SQSTM1* gene.

**Contributors** First draft of the manuscript: OC and SHR; study concept and design: SHR; obtaining funding: SHR MP and WDF; trial management during study setup and recruitment: LF, KG and SHR; genetic analysis and training of research staff in genetic counselling: MP and RC; Development of statistical analysis plan and sample size calculations: SCL and CK; design and maintenance of study database: AW; participant recruitment and study assessments: SHR, LF, KG, LRR, PLS, GH, RC, SH, JT, SY-M, MJM, RKC, WDF, LG, RN, MLB, JDP-M, J-PD, AD, GI, MDS, NG, JB, MJS, JPW, MAK, GCN, ELD, GM, AH and NLG; supervision of conduct of the trial. SHR, LF, KG, LRR, PLS, GH, RC, SH, JT, SY-M, MJM, RKC, WDF, LG, RN, MLB, JDP-M, J-PD, AD, GI, MDS, NG, JB, MJS, JPW, MAK, GCN, ELD, GDM, AH, NLG, MB, GM, KC, DW and RGGR. All authors commented on and revised the manuscript for intellectual content and approved the final version of the manuscript.

**Funding** The study was funded by the Medical Research Council (UK) (Reference number 85281) and in part by Arthritis Research UK (Reference number 18163). The zoledronic acid and placebo infusions were kindly donated by Novartis Pharmaceuticals.

**Competing interests** None declared.

**Patient consent for publication** Not required.

**Ethics approval** Ethical approval was granted by the Fife and Forth Valley Research Ethics Committee on 22 December 2008 (reference number: 08/S0501/84). The study was also approved by local research ethics committees of all participating centres outside the UK and the medicines regulatory agencies in all participating countries.

**Provenance and peer review** Not commissioned; externally peer reviewed.

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
