## [Reviewer comments · BMJ Open]

ARTICLE DETAILS

TITLE (PROVISIONAL)	Zoledronate in the prevention of Paget's (ZiPP): Protocol for a randomised trial of genetic testing and targeted Zoledronic acid therapy to prevent SQSTM1-mediated Paget's disease of bone.
AUTHORS	Cronin, Owen; Forsyth, Laura; Goodman, Kirsteen; Lewis, Steff; Keerie, Catriona; Walker, Allan; Porteous, Mary; Cetnarskyj, Roseanne; Ranganath, Lakshminarayan; Selby, Peter; Hampson, Geeta; Chandra, Rama; Ho, Shu; Tobias, Jon; Young-Min, Steven; McKenna, Malachi; Crowley, Rachel; Fraser, William; Gennari, Luigi; Nuti, Ranuccio; Brandi, Maria Luisa; Del Pino-Montes, Javier; Devogelaer, Jean-Pierre; DURNEZ, Anne; Isaia, Giancarlo; Di Stefano, Marco; Guanabens, Nuria; Blanch, Josep; Seibel, Markus; Walsh, John; Kotowicz, Mark; Nicholson, Geoffrey; Duncan, Emma; Major, Gabor; Horne, Anne; Gilchrist, Nigel; Boers, Maarten; Murray, Gordon; Charnock, Keith; Wilkinson, Diana; Russell, Graham; Ralston, Stuart

VERSION 1 – REVIEW

REVIEWER	Stergios A. Polyzos First Department of Pharmacology, Faculty of Medicine, Aristotle University of Thessaloniki, Thessaloniki, Greece
REVIEW RETURNED	22-Apr-2019

GENERAL COMMENTS	In this protocol, Cronin et al. aim to evaluate zoledronate as preventive management for individuals carrying SQSTM1 mutations, thereby being susceptible for Paget's disease of bone (PDB). The main aim is the preventive value of zoledronate in terms of new bone lesions in bone scans. Generally, it is a well-written protocol and the study is of considerable interest; however, there are some points of concern, regarding this manuscript: Major points of concern 1. A main consideration is the long period between the approval of the study by the ethics committee (2008) and the publishing of its protocol. Furthermore, the recruitment closed in 2015, therefore, it is not an a priori protocol publication.2. Some data on the actual numbers of patients recruited are provided, which is not usual when a protocol is published.3. The measurement of urine NTx may be avoided; it does not offer much more than CTX alone.4. Anxiety and depression may be evaluated by another validated questionnaire, more suitable for non-clinical individuals than the Hospital Anxiety and Depression Questionnaire (HADS).5. Page 13, lines 10-11: There is an inconsistency since it is reported that women at pregnancy or breastfeeding will be
--

	excluded from the performance of bone scan, whereas pregnancy or breastfeeding was exclusion criteria (page 11, line 32 and table 2). Minor points of concern  1. Page 7, lines 18-22: This may be rephrased, aiming to a better description. 2. Page 8, line 53: It seems that something is missing after “and”. 3. Page 11, lines 34-35: The robust form of contraception should be specifically reported, as well as the type of follow-up to assure the avoidance of pregnancy. 4. Page 12, lines 57-59: What is IMP? 5. All abbreviations should be defined the first time they appear in-text. 6. There are a few typos.
REVIEWER	Brunetti Giacomina University of Bari, Italy
REVIEW RETURNED	01-Jul-2019
GENERAL COMMENTS	The paper is interesting and well written. I just recommend to describe in the introduction the other mutations associated to Paget's disease, together with their incidence. These will explain the choice to include only patients with SQSTM1 mutations in the protocol.

VERSION 1 – AUTHOR RESPONSE

Reviewer: 1

Reviewer Name: Stergios A. Polyzos

Institution and Country: First Department of Pharmacology, Faculty of Medicine, Aristotle University of Thessaloniki, Thessaloniki, Greece

Please state any competing interests or state 'None declared': None declared

Please leave your comments for the authors below

In this protocol, Cronin et al. aim to evaluate zoledronate as preventive management for individuals carrying SQSTM1 mutations, thereby being susceptible for Paget's disease of bone (PDB). The main aim is the preventive value of zoledronate in terms of new bone lesions in bone scans. Generally, it is a well-written protocol and the study is of considerable interest; however, there are some points of concern, regarding this manuscript:

Major points of concern

1. A main consideration is the long period between the approval of the study by the ethics committee (2008) and the publishing of its protocol. Furthermore, the recruitment closed in 2015, therefore, it is not an a priori protocol publication.

Response: Thanks for this comment. While we acknowledge that submission of the protocol for publication has come several years after the initial ethical approval, we contacted the editors of BMJ Open who advised that it was permissible to publish a protocol at this stage. We believe that publication of the protocol is important since it will provide a useful overview of the methodology of the ZIPP study containing operational details that might not be possible to include in the paper which will report upon the final results of this unique trial.

2. Some data on the actual numbers of patients recruited are provided, which is not usual when a protocol is published.

Thanks for this comment. Please see response to comment 1. Although it would be possible for us to remove details of the actual number of participants recruited we thought it would be appropriate to leave this information in since it is available. We would be happy to remove this data should the editors think that would be advisable.

3. The measurement of urine NTx may be avoided; it does not offer much more than CTX alone.

Thanks for this comment. We respectfully disagree with the reviewer. The role of biochemical markers in the diagnosis of Paget's disease has recently reviewed as part of a clinical guideline published in the JBMR (reference 14 in revised manuscript). In that paper uNTX was found to have better discriminative value at identifying people with PDB in cross-sectional studies than serum CTX (raised uNTX values in more than 90% of subjects as compared with about 60% for CTX. In addition the correlation between uNTX and scintigraphic extent of PDB in a systematic review by Al-Nofal was 0.583 [95% CI; n 0.324 to 0.761] compared with 0.796 [0.702-0.862]. Although one might argue that uNTX should be used in preference to sCTX, its relevant to point out that this information wasn't available at the time the study was designed and in any case the data are derived from patients with established PDB and so we think it is reasonable to measure both markers.

Changes to paper: We have added the following passage to page 7. "The preferred markers of bone resorption are urinary NTX and serum CTX. These have been found to be elevated in patients with PDB in case control studies and to correlate with the extent of bone lesions as determined by scintigraphy in PDB (20). The preferred markers of bone formation will be PINP and BSAP since both have been shown to be superior to total ALP at detecting PDB in case control studies (17). "

4. Anxiety and depression may be evaluated by another validated questionnaire, more suitable for non-clinical individuals than the Hospital Anxiety and Depression Questionnaire (HADS).

Thank you for this comment. We acknowledge that there are other validated questionnaires that might be used to evaluate anxiety and depression. The reason for choosing HADS is that it has been extensively validate in many different countries and settings and in the UK is one of the questionnaires recommended by NICE for assessment of anxiety and depression.

Changes to paper: We have added a justification for using HADS on page information to page 8 of the revised manuscript as follows.

"This questionnaire was chosen since it is quick and simple to administer and has it has been extensively validated in many different countries and settings (25). "

5. Page 13, lines 10-11: There is an inconsistency since it is reported that women at pregnancy or breastfeeding will be excluded from the performance of bone scan, whereas pregnancy or breastfeeding was exclusion criteria (page 11, line 32 and table 2).

Thank you for highlighting this. We have amended table 3 to indicate that pregnancy or lactation at the time of randomisation would be an exclusion. We also added a section of text on page 9 to clarify the situation with regard to bone scanning and x-rays should a women become pregnancy or be lactating during the study

"Women who are pregnant or lactating at the time of randomisation will be excluded. In the event that a woman becomes pregnant or is lactating during the study, bone scanning and x-rays will not be performed until the patient is no longer pregnant and has ceased lactating. "

Minor points of concern

1. Page 7, lines 18-22: This may be rephrased, aiming to a better description.
2. Page 8, line 53: It seems that something is missing after "and".
3. Page 11, lines 34-35: The robust form of contraception should be specifically reported, as well as the type of follow-up to assure the avoidance of pregnancy.

Thank you. We have now rephrased the wording to address points 1 and 2 above and we now clarify the acceptable forms of contraception in the protocol on page 9 as follows:

"Women of childbearing potential were permitted to take part provided that they agreed to practice a medically robust form of contraception before and for at least 12 months after the ZA infusion (an intra-uterine device, a barrier method with spermicide, condoms, subdermal implant or oral contraceptive). "

4. Page 12, lines 57-59: What is IMP?

IMP stands for investigational medicinal product. We now include this where it first appears in the text.

5. All abbreviations should be defined the first time they appear in-text.

Thank you. We have now double-checked this and all abbreviations are defined where they first appear in the main text.

6. There are a few typos.

Thank you. We have reviewed the text thoroughly and believe that all typos have been corrected.

Reviewer: 2

Reviewer Name: Brunetti Giacomina

Institution and Country: University of Bari, Italy

Please state any competing interests or state 'None declared': None declared

Please leave your comments for the authors below

The paper is interesting and well written.

I just recommend to describe in the introduction the other mutations associated to Paget's disease, together with their incidence. These will explain the choice to include only patients with SQSTM1 mutations in the protocol.

Thank you for your comments.

Changes to paper: We now include references to recent papers that highlight other mutations associated with Paget's disease as follows on page 5.

"Many genetic variants have been identified that predispose to PDB and related syndromes (1, 2) but mutations in sequestosome-1 (SQSTM1) are the most important predisposing factor, occurring in up to 50% of patients with a family history of PDB and up to 10% of people who are unaware of having a family history (3-8).

VERSION 2 – REVIEW

REVIEWER	Stergios A. Polyzos First Department of Pharmacology, School of Medicine, Aristotle University of Thessaloniki, Thessaloniki, Greece
REVIEW RETURNED	31-Jul-2019
GENERAL COMMENTS	The authors responded sufficiently to the comments.